# Empirical Bayes factors for common hypothesis tests

**Frank Dudbridge** *

Department of Population Health Sciences, University of Leicester, Leicester, United Kingdom

* frank.dudbridge@leicester.ac.uk

## Abstract

Bayes factors for composite hypotheses have difficulty in encoding vague prior knowledge, as improper priors cannot be used and objective priors may be subjectively unreasonable. To address these issues I revisit the posterior Bayes factor, in which the posterior distribution from the data at hand is re-used in the Bayes factor for the same data. I argue that this is biased when calibrated against proper Bayes factors, but propose adjustments to allow interpretation on the same scale. In the important case of a regular normal model, the bias in log scale is half the number of parameters. The resulting empirical Bayes factor is closely related to the widely applicable information criterion. I develop test-based empirical Bayes factors for several standard tests and propose an extension to multiple testing closely related to the optimal discovery procedure. When only a $P$-value is available, an approximate empirical Bayes factor is $10p$. I propose interpreting the strength of Bayes factors on a logarithmic scale with base 3.73, reflecting the sharpest distinction between weaker and stronger belief. This provides an objective framework for interpreting statistical evidence, and realises a Bayesian/frequentist compromise.

**Data Availability Statement:** An R package implementing the calculations in this paper is available from https://github.com/DudbridgeLab/ebf/.

## Introduction

Bayes factors contrast the statistical evidence for two hypotheses, updating their relative plausibility in the light of data [1]. In the recent debates about testing, they offer an approach that avoids some of the difficulties with $P$-values [2–5]. However it is well known that Bayes factors for composite hypotheses are sensitive to the prior distribution of parameters [6, 7], and because improper priors cannot be used, it is not obvious how to encode vague prior knowledge [8–10]. For these reasons, Bayes factors are generally used with subjective, informative priors, or with objective priors that represent concepts of minimal information [1, 11–15].

Subjective priors are logically sound but perennially open to criticism; they may be supplemented by sensitivity analysis, but that may be tedious and distracting in routine applications. Objective priors offer the prospect of standardised analyses; however, they do not reflect a genuine state of knowledge and may be at odds with reasonable assumptions about the subject matter [8, 16, 17]. They may also be at odds with future studies that use knowledge learned from the data at hand, playing into concerns about reproducibility [18]. These issues have

**Funding:** This work was supported by the Medical Research Council https://www.ukri.org/councils/mrc/ (MR/S037055/1 to FD). The funders had no role in study design, data collection and analysis, decision to publish, or preparation of the manuscript.

**Competing interests:** The authors have declared that no competing interests exist.

inhibited the use of Bayes factors with the common, textbook statistics upon which much applied research depends.

This paper aims to bridge this gap by constructing objective Bayes factors with priors that are subjectively reasonable: that is, they can be considered compatible with the subject matter. I take the view that from a position of complete ignorance, a reasonable prior can only be inferred from the data at hand. One approach is to use part of the data to learn the prior and the remainder to evaluate the Bayes factor, an idea developed in the fractional and intrinsic Bayes factors [19, 20]. However, there is no good general rule for dividing the data, and different allocations may suit different problems. On the other hand, the posterior Bayes factor [21] infers the prior from the entire data and then evaluates the Bayes factor in the same data. This approach was criticised for using the data twice, but that criticism only holds sway if the posterior Bayes factor is held to an agreed calibration; it is otherwise unclear how inferences are distorted (see discussion and rejoinder of Aitkin [21]). The proposal is to compare the posterior Bayes factor to the Bayes factor from an independently supplied prior with the same information. It will be seen that the posterior Bayes factor tends to over-estimate the evidence compared to such a proper Bayes factor, and it is therefore biased in that respect.

Here I develop the posterior Bayes factor in two ways. Firstly I correct this bias to give empirical Bayes factors (EBFs) that are interpretable on the same scale as ordinary Bayes factors. Secondly I calculate EBFs from summary test statistics, giving test-based Bayes factors [22]. This allows standard routine analyses to proceed as usual, with the EBF calculated at the same stage as a *P*-value.

Similar ideas occur in posterior predictive approaches to model selection, including Bernardo's Bayesian reference criterion (BRC) [23] and Watanabe's widely applicable information criterion (WAIC) [24, 25]. Indeed for regular normal models, the EBF is equivalent to the difference in WAIC under a vague prior. The EBF differs in explicitly seeking a Bayes factor interpretation. The focus on computing EBFs from summary statistics, and in multiple testing applications, also leads to differences from BRC and WAIC.

The EBF is empirical Bayes in that it is framed as a Bayesian quantity with prior estimated from the data. Empirical Bayes ideas have been applied to multiple testing for some time [26], but with the focus on posterior probabilities rather than the Bayes factor. Storey's optimal discovery procedure [27] can be interpreted as an EBF, although it was developed as a frequentist statistic and does not explicitly estimate a prior distribution. The Bayesian decision procedure of Guindani et al. [28] also resembles an EBF. Closest to the approach developed here is work by Morisawa et al. [29], who constructed EBFs for multiple tests from a non-parametric estimate of the prior distribution. But they did not interpret their EBF beyond comparing its ranking of associations to that from *P*-values. Thus, while concepts related to EBFs have been in the multiple testing literature for some time, there has been little effort to interpret them in the same way as ordinary Bayes factors.

I am emphasising the Bayes factor interpretation, but it is unclear why this would be desirable. Although Bayes factors update prior odds to posterior odds, it is uncommon for either prior or posterior odds to be appraised precisely. More usually the strength of evidence is assessed qualitatively by the order of magnitude of the Bayes factor. But proposed interpretations with base 10 [9, 30, 31], natural logs [1] and base 2 [32] are based on subjective experiences, undermining the aim of an objective Bayes factor. Here I propose interpreting Bayes factors with base 3.73, using mathematical arguments flowing from Bayes' theorem. Such an objective scale of interpretation is lacking from other accounts of evidence [33, 34], and gives a rationale for the Bayes factor interpretation separate from more philosophical arguments. With this scale, the EBF yields a surprising re-interpretation of classical significance testing, realising a compromise between Bayesian and frequentist paradigms.

## Single tests

The Bayes factor in favour of hypothesis $H_0$ over $H_1$, in light of data $x$, is

$$\frac{\Pr(x|H_0)}{\Pr(x|H_1)}$$

It updates prior probability to posterior probability according to the odds version of Bayes' theorem,

$$\frac{\Pr(H_0|x)}{\Pr(H_1|x)} = \frac{\Pr(x|H_0)}{\Pr(x|H_1)}\frac{\Pr(H_0)}{\Pr(H_1)}$$

Suppressing the subscript, when $H$ describes a set $\Theta_H$ of parameter values in a model for $x$,

$$\Pr(x|H) = \int_{\Theta_H} f(x|\theta)\pi_H(\theta)d\theta$$

where $f(x|\theta)$ is the density of $x$ given $\theta$, and $\pi_H(\theta)$ is the prior density of $\theta \in \Theta_H$ under $H$ (similar definitions hold with probabilities for discrete $x$ or $\theta$). Various terms are used for $\Pr(x|H)$; here it will be called the (prior) marginal likelihood.

Let $\pi_H(\theta|x)$ denote a posterior density for $\theta \in \Theta_H$ in light of $x$. Take an unrestricted prior $\pi(\theta)$ truncated to $\Theta_H$ so that $\pi_H(\theta|x)$ is similarly the truncated $\pi(\theta|x)$. The *posterior marginal likelihood* is given by

$$M_H(x) = \frac{\int_{\Theta_H} f(x|\theta)\pi(\theta|x)d\theta}{\int_{\Theta_H}\pi(\theta|x)d\theta} \tag{1}$$

equal to the posterior predictive density of $x$ itself. The posterior marginal likelihood may be biased, in the sense of being systematically different from a marginal likelihood with the prior for $\theta$ supplied independently of $X$. To fix this idea, consider the marginal likelihood using, as the prior, the posterior distribution from independent replicate data. Such data are understood in the frequentist sense as exchangeable with the data at hand, in particular having the same sample size. Given replicate data $y$, the bias in $\log M_H(X)$ is

$$b_H(y) = E_X\left[\log M_H(X) - \log\frac{\int_{\Theta_H} f(X|\theta)\pi(\theta|y)d\theta}{\int_{\Theta_H}\pi(\theta|y)d\theta}\right]$$

The *expected bias* $E_Y b_H(Y)$ is taken over all replicate data, and the bias-corrected posterior marginal likelihood is

$$M_H(x)\exp(-E_Y b_H(Y))$$

The EBF is the ratio of the bias-corrected posterior marginal likelihoods,

$$EBF_{01}(x) = \frac{M_{H_0}(x)\exp(-E_Y b_{H_0}(Y))}{M_{H_1}(x)\exp(-E_Y b_{H_1}(Y))} \tag{2}$$

Calculation of the bias requires a distribution for $X$ and $Y$. I will use the joint prior predictive distribution, $\Pr(x,y) = \int f(x|\theta)f(y|\theta)\pi(\theta)d\theta$ where $\theta$ is now not restricted to $\Theta_H$. In other words I consider the evidence for $H$ under all data $X$, $Y$ that may arise under model $f(\cdot)$

whether or not $H$ is true. The expected bias is then explicitly

$$E_Y b_H(Y) = \iint \left[ \int f(x|\theta)f(y|\theta)\pi(\theta)d\theta \{\log \int_{\Theta_H} f(x|\theta^*)\pi(\theta^*|x)d\theta^* - \log \int_{\Theta_H} f(x|\theta^*)\pi(\theta^*|y)d\theta^* \} \right] dy \, dx \quad (3)$$

where the outer integrals are over the entire domains of $f(\cdot)$ and $\pi(\cdot)$ respectively. Changing the order of integration gives

$$E_Y b_H(Y) = \int \left[ \iint f(x|\theta)f(y|\theta) \{\log \int_{\Theta_H} f(x|\theta^*)\pi(\theta^*|x)d\theta^* - \log \int_{\Theta_H} f(x|\theta^*)\pi(\theta^*|y)d\theta^* \} dy \, dx \right] \pi(\theta)d\theta \quad (4)$$

This will be useful when the integral in $x$ and $y$ does not depend on $\theta$, in which case the outer integral over $\pi(\theta)$ may be discarded. This occurs in most of the following subsections.

The prior $\pi(\theta)$ enters the EBF both through the marginal likelihood and the bias correction. The approach to identifying priors will aim both to express vague information about $\theta$ and to produce a reasonable prior predictive distribution for the bias correction [35, 36]. This position is especially relevant for the binomial probability.

The EBF may be viewed as a bias-corrected posterior Bayes factor, or as the expected Bayes factor (in log scale) if the prior were provided from independent replicate data. Another interpretation is the expected Bayes factor (in log scale) if the posterior distribution from the data at hand were taken as the prior in independent replicate data. This reflects what might actually be done in practice, and in that respect the EBF can be viewed as a reproducible measure of evidence.

## Normal theory

Let the data be summarised by a single vector observation $X$ such that the likelihood for the mean is $\phi_d(x; \mu, \Sigma)$, the multivariate normal density of dimension $d$ with mean vector $\mu$, and the variance-covariance matrix $\Sigma$ assumed known and of full rank. Here $X$ may be any quantity with an assumed normal distribution, especially large sample parameter estimates. With an improper flat prior for $\mu$, the posterior is $\pi(\mu|x; \Sigma) = \phi_d(\mu; x, \Sigma)$. The numerator of the posterior marginal likelihood is

$$\int_{\Theta_H} \phi_d(x; \mu, \Sigma)\phi_d(\mu; x, \Sigma)d\mu = (4\pi|\Sigma|)^{-\frac{1}{2}} \int_{\Theta_H} \phi_d(\mu; x, \Sigma/2)d\mu \quad (5)$$

For the unrestricted hypothesis $H{:}\mu{\in}\mathbb{R}^d$, the posterior marginal likelihood is just $(4\pi|\Sigma|)^{-\frac{1}{2}} = \phi_d(x; x, 2\Sigma)$ and the posterior predictive density of $y$ is $\phi_d(y; x, 2\Sigma)$. Then the bias is

$$b_H(y) = E_X[\log \phi_d(X; X, 2\Sigma) - \log \phi_d(X; y, 2\Sigma)]$$

$$= \frac{1}{2} E_X \left[ (X - y)' \frac{\Sigma^{-1}}{2} (X - y) \right]$$

$$= \frac{1}{2} E_X \left[ tr \left\{ (X - y)' \frac{\Sigma^{-1}}{2} (X - y) \right\} \right]$$

$$= \frac{1}{2} E_X \left[ tr \left\{ \frac{\boldsymbol{\Sigma}^{-1}}{2} (\boldsymbol{X} - \boldsymbol{y})(\boldsymbol{X} - \boldsymbol{y})' \right\} \right]$$

$$= \frac{1}{2} tr \left[ \frac{\boldsymbol{\Sigma}^{-1}}{2} E_X \{ (\boldsymbol{X} - \boldsymbol{y})(\boldsymbol{X} - \boldsymbol{y})' \} \right]$$

The expected bias is

$$E_Y b_H(\boldsymbol{Y}) = \frac{1}{2} tr \left[ \frac{\boldsymbol{\Sigma}^{-1}}{2} E_X E_Y \{ (\boldsymbol{X} - \boldsymbol{Y})(\boldsymbol{X} - \boldsymbol{Y})' \} \right] = \frac{d}{2} \tag{6}$$

In a regular normal model, the expected bias is half the dimension, irrespective of $\boldsymbol{\mu}$. The bias-corrected posterior marginal likelihood is

$$M_H(x) \exp(-E_Y b_H(\boldsymbol{Y})) = \phi_d(\boldsymbol{x}; \boldsymbol{x}, 2\boldsymbol{\Sigma}) \exp(-d/2)$$

$$= \phi_d(\boldsymbol{x}; \boldsymbol{x}, \boldsymbol{\Sigma}) 2^{-\frac{d}{2}} \exp(-d/2)$$

On the deviance scale this is

$$-2 \log[M_H(x) \exp(-E_Y b_H(\boldsymbol{Y}))] = -2 \log \phi_d(\boldsymbol{x}; \boldsymbol{x}, \boldsymbol{\Sigma}) + d(1 + \log(2)) \tag{7}$$

that is, minus twice the maximised log-likelihood penalised by the number of parameters times 1+log(2) = 1.69. This resembles the Akaike information criterion (AIC), but with a weaker penalty, and in fact it is equal to WAIC in the case of a regular normal model with asymptotically flat prior [24, 25]. This is unsurprising since the posterior marginal likelihood is also the posterior predictive density of $x$ itself, which the WAIC adjusts to the posterior predictive density in new data. Therefore, in model selection the penalty of 1.69$d$ gives a WAIC-like criterion interpretable on the Bayes factor scale. However, the focus on test-based EBFs leads to differences from WAIC in general.

Now consider a directional hypothesis, for example scalar $H{:}\mu{>}0$. The posterior distribution is truncated to $(0, \infty]$

$$\pi(\mu|x, \sigma^2) = \frac{\phi(\mu; x, \sigma^2)}{\int_0^\infty \phi(\mu; x, \sigma^2)}$$

and the expected bias is

$$E_Y b_H(Y) = E_X E_Y [\log \int_0^\infty \phi(X; \mu, \sigma^2) \phi(\mu; X, \sigma^2) d\mu - \log \int_0^\infty \phi(X; \mu, \sigma^2) \phi(\mu; Y, \sigma^2) d\mu]$$

which now depends on the prior predictive distribution of $X$ and $Y$. Note that for sufficiently large $X$, $\phi(X; \mu, \sigma^2)$ is arbitrarily close to zero for $\mu{<}0$ and the integrals in the expected bias approach those for the unrestricted hypothesis. In the limit of a flat prior on $\mu$ this occurs with probability $\frac{1}{2}$. Similarly with probability $\frac{1}{2}$, $-X$ is large enough that $\phi(X; \mu, \sigma^2)$ is arbitrarily close to zero for $\mu{>}0$ and the integrals then approach zero. The expected bias is thus one half of that for the unrestricted hypothesis, $E_Y b_H(Y) = \frac{1}{4}$.

In some one-sided problems, values of $\mu{<}0$ (say) are impossible by definition. In these cases the above argument for $-X$ does not apply and the expected bias is $\frac{1}{2}$ as for the unrestricted hypothesis.

In the multivariate case the expected bias can be taken as $\frac{d_1 + 2d_2}{4}$, where $d_1$ is the number of one-sided hypotheses and $d_2$ is the number of two-sided hypotheses for the elements of $\boldsymbol{\mu}$. Note that there is no bias for finite interval or point hypotheses.

From Eqs (2), (5) and (6) it is easy to construct EBFs for two-sided, one-sided and interval hypotheses about a scalar mean. For the two-sided test of $H_0: \mu = 0$ versus $H_1: \mu \neq 0$, the EBF in favour of $H_0$ is

$$EBF_{01} = \sqrt{2} \exp\left(-\frac{1}{2}(z^2 - 1)\right) \tag{8}$$

where $z = x/\sigma$.

For $H_0: \mu = 0$ versus the one-sided alternative $H_1: \mu > 0$ the EBF is

$$EBF_{01} = \frac{\Phi(z)\sqrt{2}}{\Phi(z\sqrt{2})} \exp\left(-\frac{1}{2}\left(z^2 - \frac{1}{2}\right)\right) \tag{9}$$

assuming that $\mu < 0$ is also possible. If $\mu < 0$ is impossible then the EBF is

$$EBF_{01} = \frac{\Phi(z)\sqrt{2}}{\Phi(z\sqrt{2})} \exp\left(-\frac{1}{2}(z^2 - 1)\right) \tag{10}$$

For $H_0: \mu < 0$ versus $H_1: \mu > 0$, the EBF is

$$EBF_{01} = \frac{\Phi(-z\sqrt{2})}{\Phi(z\sqrt{2})} \frac{\Phi(z)}{\Phi(-z)} \tag{11}$$

Some implications of these results are discussed below in the Case studies.

The multivariate version of Eq (8) is

$$EBF_{01} = 2^{\frac{d}{2}} \exp\left(-\frac{1}{2}(z^2 - d)\right) \tag{12}$$

where $z^2 = \boldsymbol{x}^T \Sigma^{-1} \boldsymbol{x}$. This corresponds to a $\chi^2$ test when $Z^2 \sim \chi_d^2$ under $H_0$ and has a non-central $\chi^2$ distribution under $H_1$.

### *t*-tests

In the following subsections I consider scalar quantities in some commonly used tests. The principles for constructing their EBFs may be applied to other tests not treated here.

Let $X$ and $S$ be observed such that for some $\mu$, $(X - \mu)/S$ is distributed as $t$ with $v$ degrees of freedom. The location $\mu$ is the parameter of interest, with likelihood $t_v(x; \mu, s) = s^{-1} t_v((x - \mu)/s)$.

With a flat prior for $\mu$, the posterior is $\pi(\mu | x, s) = t_v(\mu; x, s)$. The numerator of the posterior marginal likelihood is

$$\int_{\Theta_H} f(x | \mu, s) \pi(\mu | x, s) d\mu = \int_{\Theta_H} \left[ \frac{\Gamma\left(\frac{v+1}{2}\right)}{s\sqrt{v\pi}\Gamma\left(\frac{v}{2}\right)} \left(1 + \frac{1}{v}\left(\frac{x - \mu}{s}\right)^2\right)^{\frac{-v-1}{2}} \right]^2 d\mu$$

$$= \frac{\Gamma\left(\frac{v+1}{2}\right)^2}{s^2 v\pi \Gamma\left(\frac{v}{2}\right)^2} \int_{\Theta_H} \left(1 + \frac{1}{2v+1}\left(\frac{x - \mu}{s}\right)^2 \frac{2v+1}{v}\right)^{\frac{-2v-2}{2}} d\mu$$

**Table 1. Expected bias $E_Y b_H(Y)$ of the log posterior marginal likelihood for the $t$ distribution with $v$ degrees of freedom.**

| $v$ | 1 | 2 | 3 | 4 | 5 | 6 | 7 | 8 | 9 | 10 |
|---|---|---|---|---|---|---|---|---|---|---|
| $E_Y b_H(Y)$ | 1.39 | 0.860 | 0.710 | 0.644 | 0.608 | 0.586 | 0.571 | 0.560 | 0.552 | 0.546 |

$$= \frac{\Gamma\left(\frac{v+1}{2}\right)^2 \Gamma\left(v+\frac{1}{2}\right)}{s\sqrt{v\pi}\,\Gamma\left(\frac{v}{2}\right)^2 \Gamma(v+1)} \int_{\Theta_H} t_{2v+1}\left(\mu; x, s\sqrt{\frac{v}{2v+1}}\right) d\mu \qquad (13)$$

For the expected bias,

$$E_Y b_H(Y) = \int_{-\infty}^{\infty}\int_{\infty}^{\infty} t_v(x;\mu,s) t_v(y;\mu,s)$$

$$\times \left\{ \log \int_{\Theta_H} t_v(x;\mu^*,s) t_v(\mu^*;x,s) d\mu^* - \log \int_{\Theta_H} t_v(x;\mu^*,s) t_v(\mu^*;y,s) d\mu^* \right\} dy\, dx$$

$$= \int_{-\infty}^{\infty}\int_{\infty}^{\infty} t_v(x) t_v(y)$$

$$\times \left\{ \log \int_{\Theta_H} t_v(x-\mu^*) t_v(\mu^*-x) d\mu^* - \log \int_{\Theta_H} t_v(x-\mu^*) t_v(\mu^*-y) d\mu^* \right\} dy\, dx \quad (14)$$

irrespective of $\mu$. Numerical integration for $v = 1,\ldots,10$ is shown in Table 1. The bias is greater than for the normal distribution, but decreases to the asymptotic value of $\frac{1}{2}$ as $v$ increases; for $v = 30$, the expected bias is 0.513. As with the normal distribution, the bias is halved for directional hypotheses and vanishes for interval hypotheses.

This formulation accommodates the most common applications of $t$-tests. For tests of the mean, $X$ is the sample mean (or for two samples, difference in means) and $S$ its estimated standard error. Similarly, for Wald tests of regression parameters, $X$ is the parameter estimate and $S$ its estimated standard error. Two-sided tests of $H_0$:$\mu = \mu_0$ have $\Theta_{H_1} = [-\infty, \infty]$ and one-sided tests have $\Theta_{H_1} = [-\infty, \mu_0]$ or $\Theta_{H_1} = [\mu_0, \infty]$ as appropriate, with the integration in Eq 13 calculated from the cumulative distribution of $t$.

### Binomial probability

Consider a fixed number $n$ of Bernoulli trials with $x$ successes. The likelihood for the probability $p$ is Binomial $(n, p)$. Take the usual $Beta(\alpha, \alpha)$ prior for $p$, with posterior $Beta(x+\alpha, n-x+\alpha)$; I discuss the choice of $\alpha$ below. The numerator of the posterior marginal likelihood is

$$\int_{\Theta_H} f(x|p,n)\pi(p|x,n)dp = \binom{n}{x} \frac{B(2x+\alpha, 2(n-x)+\alpha)}{B(x+\alpha, n-x+\alpha)} \int_{\Theta_H} f_B(p; 2x+\alpha, 2(n-x)+\alpha)dp \quad (15)$$

where $f_B(p; \alpha, \beta)$ denotes the $Beta(\alpha, \beta)$ density at $p$. For the expected bias we have

$$E_Y b_H(Y) = \sum_{x=0}^{n}\sum_{y=0}^{n} \Pr(x,y) D_H(x,y,n,\alpha)$$

where

$$\Pr(x,y) = \binom{n}{x}\binom{n}{y} \frac{B(x+y+\alpha, 2n-x-y+\alpha)}{B(\alpha,\alpha)}$$

**Table 2. Expected bias $E_Y b(Y)$ of the log posterior marginal likelihood of the binomial distribution with $n$ trials, with uniform prior.**

| $n$ | 1 | 2 | 3 | 4 | 5 | 6 | 7 | 8 | 9 | 10 |
|---|---|---|---|---|---|---|---|---|---|---|
| $\Theta_H = [0,1]$ | 0.231 | 0.316 | 0.360 | 0.387 | 0.405 | 0.418 | 0.428 | 0.436 | 0.442 | 0.447 |
| $\Theta_H = [0,0.5]$ | 0.093 | 0.133 | 0.157 | 0.172 | 0.183 | 0.191 | 0.198 | 0.203 | 0.207 | 0.210 |

and

$$D_H(x,y,n,\alpha) = \begin{array}{l} \log\left[\binom{n}{x} B(2x+\alpha, 2(n-x)+\alpha) \int_{\Theta_H} f_B(p; 2x+\alpha, 2(n-x)+\alpha) dp\right] - \\[2ex] \log\left[\binom{n}{y} B(x+y+\alpha, 2n-x-y+\alpha) \int_{\Theta_H} f_B(p; x+y+\alpha, 2n-x-y+\alpha) dp\right] \end{array} \tag{16}$$

There are good arguments for different values of $\alpha$ [37, 38]. Here I prefer the uniform prior as it is consistent with a uniform prior predictive $\Pr(x) = (n+1)^{-1}$ [36, 39]. This is desirable because the expected bias explicitly depends on $\Pr(x)$, for which the uniform distribution is a natural non-parametric choice. The expected bias for $n = 1,\ldots,10$, calculated from Eq (16) for $\Theta_H = [0,1]$ and $\Theta_H = [0,0.5]$ is shown in Table 2.

As $n$ increases, the expected bias approaches the normal theory values of 0.5 for the full interval and 0.25 for the half interval. The expected bias for $\Theta_H = [0.5,1]$ is the same as for $\Theta_H = [0,0.5]$ so that in a comparison of $H_0$:$p \le 0.5$ to $H_1$:$p > 0.5$, the bias cancels.

If instead the number of successes is fixed at $x$ and the total number of trials $n$ is random, similar calculations yield EBFs for the negative binomial distribution. In general, for given $x$ and $n$ the EBF is different for binomial and negative binomial models. This dependence on the sampling model is a violation of the likelihood principle, but without a sampling model one cannot identify a bias in the posterior Bayes factor. In practice, if the sampling process is uncertain one may use Bayesian model averaging to reduce the dependence on the sampling model. In its most objective form this simply calculates the bias-corrected posterior marginal likelihoods for each possible model and then takes their arithmetic mean.

## *F*-tests

Let $X$ be observed such that for some $r>0$, $rX$ is distributed as $F$ with $(v_1, v_2)$ degrees of freedom. The likelihood for $r$ is $r f_{v_1,v_2}(rx)$. Take the usual improper prior for a scale parameter $\pi(r) \propto r^{-1}$, equivalent to a flat prior on $\log r$. The posterior density is $\pi(r|x) = x f_{v_1,v_2}(rx)$. The numerator of the posterior marginal likelihood is

$$\int_{\Theta_H} f(x|r)\pi(r|x)dr = \int_{\Theta_H} r x f_{v_1,v_2}(rx)^2 dr$$

$$= \int_{\Theta_H} r x \frac{v_1^{v_1} v_2^{v_2} (rx)^{v_1-2}}{(v_1 rx + v_2)^{v_1+v_2} B\left(\frac{v_1}{2},\frac{v_2}{2}\right)^2} dr$$

$$= \frac{B(v_1, v_2)}{B\left(\frac{v_1}{2},\frac{v_2}{2}\right)^2} \int_{\Theta_H} f_{2v_1, 2v_2}(rx)dr \tag{17}$$

Similar to the *t*-test, the expected bias is calculated by numerical integration.

$$E_Y b_H(Y) = \int_0^\infty \int_0^\infty r^2 f_{v_1,v_2}(rx) f_{v_1,v_2}(ry) \{ \log \int_{\Theta_H} r^* x f_{v_1,v_2}(r^*x)^2 dr^* - \log \int_{\Theta_H} r^* x f_{v_1,v_2}(r^*x) f_{v_1,v_2}(r^*y) dr^* \} dy\, dx$$

$$= \int_0^\infty \int_0^\infty f_{v_1,v_2}(x) f_{v_1,v_2}(y) \{ \log \int_{\Theta_H} r^* x f_{v_1,v_2}(r^*x)^2 dr^* - \log \int_{\Theta_H} r^* x f_{v_1,v_2}(r^*x) f_{v_1,v_2}(r^*y) dr^* \} dy\, dx \quad (18)$$

irrespective of *r*. Some values are shown in Table 3.

A common application of *F*-tests is in the analysis of variance, in which the tests are one-sided. In the present notation the hypotheses are $H_0:r = 1$ versus $H_1:r<1$. The posterior marginal likelihood is then

$$M_H(x) = \frac{B(v_1, v_2) \int_0^1 f_{2v_1,2v_2}(rx) dr}{B\left(\frac{v_1}{2}, \frac{v_2}{2}\right)^2 \int_0^1 x f_{v_1,v_2}(rx) dr}$$

$$= \frac{B(v_1, v_2) F_{2v_1,2v_2}(x)}{B\left(\frac{v_1}{2}, \frac{v_2}{2}\right)^2 x F_{v_1,v_2}(x)} \quad (19)$$

where *F* denotes the cumulative distribution function. Here values of $r>1$ are implausible by construction, so following a similar argument to the normal theory the expected bias is the same as for the unrestricted hypothesis.

## *P*-values

In some situations only a *P*-value may be available, such as in non-parametric tests based on ranks. Here an EBF can be obtained by assuming that *P*-values arise from a Beta distribution. This assumption may not hold in reality, and such EBFs should be viewed as no more than a rough measure. However, the following result provides a useful rule of thumb.

Previously, Sellke et al. [40] obtained the lower bound of $-e\, p \log(p)$ for the Bayes factor when $p<e^{-1}$, assuming that the *P*-value follows a *Beta*($\alpha$, 1) distribution with $\alpha<1$. Held and Ott [41] considered a *Beta*(1,$\beta$) distribution with $\beta>1$, obtaining a smaller lower bound of $-e(1-p)\log(1-p)$. Here I adopt the latter model as it can accommodate a flat prior and yields an expected bias that is fairly constant over $\beta$.

Let *p* be a *P*-value in [0,1]. The likelihood for $\beta$ is $f_B(p; 1, \beta) = \beta(1-p)^{\beta-1}$. Under a flat prior for $H:\beta>1$ the posterior is *Gamma*(2,$-\log(1-p)$) and the posterior marginal likelihood is

$$M_H(p) = \frac{\int_1^\infty \beta^2 (1-p)^{2\beta-1} d\beta}{\int_1^\infty \beta(1-p)^\beta d\beta} \quad (20)$$

**Table 3. Expected bias of the log posterior marginal likelihood for the *F* distribution. Numerator degrees of freedom in rows, denominator degrees of freedom in columns.**

| $v_1; v_2$ | 1 | 5 | 10 | 20 | 50 |
|---|---|---|---|---|---|
| 1 | 0.609 | 0.650 | 0.670 | 0.681 | 0.688 |
| 5 | 0.650 | 0.527 | 0.526 | 0.534 | 0.542 |
| 10 | 0.670 | 0.526 | 0.513 | 0.513 | 0.518 |
| 20 | 0.681 | 0.534 | 0.513 | 0.506 | 0.507 |
| 50 | 0.688 | 0.542 | 0.518 | 0.507 | 0.503 |

Calculus gives the numerator as

$$(1-p)\int_0^{-\infty} e^b \left( \frac{b^2}{8[\log(1-p)]^3} - \frac{b}{2[\log(1-p)]^2} + \frac{1}{2\log(1-p)} \right) db = (1-p) \left[ \frac{-1}{4[\log(1-p)]^3} + \frac{1}{2[\log(1-p)]^2} - \frac{1}{2\log(1-p)} \right]$$

and the denominator as

$$(1-p)\int_0^{-\infty} e^b \left( \frac{b}{[\log(1-p)]^2} + \frac{1}{\log(1-p)} \right) db = (1-p) \left[ \frac{1}{[\log(1-p)]^2} - \frac{1}{\log(1-p)} \right]$$

For a fixed value of $\beta$ the expected bias has the form

$$E_Q b_H(Q) = \int_0^1 \int_0^1 f_B(p;1,\beta) f_B(q;1,\beta)$$

$$\times \left\{ \log \int_1^\infty f_B(p;1,\beta^*) f_G(\beta^*;2,-\log(1-p)) d\beta^* - \log \int_1^\infty f_B(q;1,\beta^*) f_G(\beta^*;2,-\log(1-p)) d\beta^* \right\} dq \, dp \quad (21)$$

The first integral in the bracket is the numerator above. The second integral is

$$\int_1^\infty \beta^{*2} (1-p)^{\beta^*} (1-q)^{\beta^*-1} d\beta^*$$

$$= (1-p) \left[ \frac{-2}{[\log((1-p)(1-q))]^3} + \frac{2}{[\log((1-p)(1-q))]^2} - \frac{1}{\log((1-p)(1-q))} \right]$$

It is found that $E_Q b_H(Q) \approx \log\left(\frac{5}{2}\right) = 0.916$ for a wide range of $\beta$, and so this value is taken as a default. Assume that under $H_0$, the $P$-value is uniformly distributed and the likelihood is 1. Then a $P$-value of 0.05 equates to an EBF of 1/2.05, compared to the Sellke et al. [40] lower bound of 1/2.45 and the Held and Ott [41] lower bound of 1/7.55.

For small $P$-values, say less than 0.1, the numerator of $M_H(p)$ is dominated by

$$\frac{-(1-p)}{4[\log(1-p)]^3} \approx \frac{1-p}{4p^3}$$

and the denominator by

$$\frac{1-p}{[\log(1-p)]^2} \approx \frac{1-p}{p^2}$$

Then with the default bias adjustment the EBF is approximated by

$$EBF_{01} \approx \frac{4p^3}{p^2} \frac{5}{2}$$

$$= 10p \quad (22)$$

Thus a small $P$-value can be converted into an EBF using a very simple rule of thumb. Indeed, with equal prior probabilities on $H_0$ and $H_1$, the corresponding posterior probability of $H_0$ is $1-(1+10p)^{-1}$. When $p = 0.05$, this probability is $\frac{1}{3}$, consistent with the lower bound of around 30% proposed by Berger and Sellke [42]. For $p = 0.005$, proposed as a revised standard for significance testing [43], the posterior probability of $H_0$ is 4.8%.

## Multiple tests

For multiple tests the principle is again to estimate a posterior distribution from the data, and construct a bias-corrected posterior marginal likelihood for each hypothesis in each test. Now the posterior distribution is estimated from the ensemble of tests, assuming exchangeability under a common random-effects distribution. This is not correcting for multiple testing in the usual sense, but is including more information in each test, potentially improving simultaneous inference in both frequentist and Bayesian paradigms [27, 28]. For simplicity I describe an EBF for scalar parameters, but the generalisation to vectors is straightforward.

Let the data be $X_i \sim F(\theta_i; \eta_i)$ independently for $i = 1, \ldots, m$, where $F$ is a known distribution, $\theta_i$ is the parameter of interest in test $i$ and $\eta_i$ the nuisance parameter assumed known. Let $\Theta$ be a common set such that the hypothesis in test $i$ is $H_{(i)}: \theta_i \in \Theta$. We want a posterior distribution for $\theta_i$ given that $H_{(i)}$ is true, but $H_{(j)}$ is not necessarily true for $j \neq i$. Let $p_{\Theta;i}$ be a common prior probability that $\theta_j \in \Theta$ for $j \neq i$. Then take the mixture of single-test posterior distributions

$$\pi(\theta_i | \boldsymbol{x}; p_{\Theta;i}) \propto \pi(\theta_i | x_i; \eta_i) + p_{\Theta;i} \sum_{j \neq i} \pi(\theta_i | x_j; \eta_j) \tag{23}$$

The posterior marginal likelihood for test $i$ is

$$M_{H_{(i)}}(x_i; \boldsymbol{x}, \boldsymbol{\eta}) = \frac{\int_\Theta f(x_i; \theta, \eta_i) \pi(\theta | x_i; \eta_i) d\theta + p_{\Theta;i} \sum_{j \neq i} \int_\Theta f(x_i; \theta, \eta_i) \pi(\theta | x_j; \eta_j) d\theta}{\int_\Theta \pi(\theta | x_i; \eta_i) d\theta + p_{\Theta;i} \sum_{j \neq i} \int_\Theta \pi(\theta | x_j; \eta_j) d\theta}$$

where $f$ is the density corresponding to $F$. The integrals in the numerator are the same as those appearing in the expected bias calculations for single tests, with $x_j$ in place of replicate data $y$.

The bias in $\log M_{H_{(i)}}(x_i; \boldsymbol{x}, \boldsymbol{\eta})$ now involves logs of sums and cannot be easily derived. The following heuristic is proposed. The first term in the numerator is adjusted by the single-test correction from the previous section, while the other terms involving independent data are left unchanged. The adjusted posterior marginal likelihood for test $i$ is thus

$$\frac{\exp(-E_{Y_i} b_{H_{(i)}}(Y_i)) \int_\Theta f(x_i; \theta, \eta_i) \pi(\theta | x_i; \eta_i) d\theta + p_{\Theta;i} \sum_{j \neq i} \int_\Theta f(x_i; \theta, \eta_i) \pi(\theta | x_j; \eta_j) d\theta}{\int_\Theta \pi(\theta | x_i; \eta_i) d\theta + p_{\Theta;i} \sum_{j \neq i} \int_\Theta \pi(\theta | x_j; \eta_j) d\theta} \tag{24}$$

This approach is related to the ODP [27]. In the present notation the ODP uses $m^{-1} \sum_j f(x_i; \theta_j, \eta_j)$ involving the true but unknown parameters $\theta_j$. In practice they must be estimated, here using posterior distributions. I use a single probability $p_{\Theta;i}$ for all tests $j \neq i$ and treat this as an input parameter, but different probabilities could be specified for each test [44]. The main conceptual differences from the ODP are that I assume $H_{(i)}$ is true for test $i$, and recognise a bias from double use of $x_i$ (Eq 24). These differences arise because I seek a Bayes factor interpretation.

When the number of tests $m$ is large, the contribution of each data point to its own test becomes negligible and $p_{\Theta;i}$ almost cancels from Eq (24). In practice the EBF is often insensitive to values of $p_{\Theta;i}$ outside a small range near 0. An example is given in the Case Studies.

A related strand of work has focussed on the posterior probability of a point null hypothesis, compared to a general alternative [26, 45]. The observations are assumed to follow a two-groups model

$$f(X) = \pi_0 f_0(X) + (1 - \pi_0) f_1(X)$$

where $f(X)$ is the marginal density of $X$, $f_i$ is the density under hypothesis $H_i$ and $\pi_0$ is the prior

probability of $H_0$. Then according to Bayes' formula the posterior probability is

$$\Pr(H_0|X) = \frac{\pi_0 f_0(X)}{f(X)}$$

Since $f$ can be estimated directly from the observations, and $f_0$ may be assumed by theory, there is a straightforward conversion of prior probability to posterior probability. The approach is even more appealing if $\pi_0$ itself can be estimated from the data [46, 47], for we then directly obtain posterior probabilities of the hypotheses, ostensibly without prior information.

In practice $\pi_0$ cannot be estimated without some prior assumption. Intuitively, in finite data a model with $\pi_0 > 0$ is almost indistinguishable from a model with $\pi_0 = 0$ and infinitesimally small departures from $H_0$. Thus a degree of subjectivity is inevitable; usually the assumption is that $\pi_0$ is close to 1. With $\pi_0$ given, the problem reduces to density estimation for the marginal $f(X)$.

In this model the Bayes factor is implicitly defined as

$$\frac{f_0(X)}{f_1(X)} = \frac{(1 - \pi_0)f_0(X)}{f(X) - \pi_0 f_0(X)}$$

Of note, $\pi_0$ is the same for all tests and appears both in the Bayes factor and as the prior probability in each test. In contrast, the EBF model for $f_1$ allows $p_{\Theta;i}$ to differ for each test and to differ from the prior probability, which does not feature in the EBF. While $p_{\Theta;i}$ can be made the same for all tests, it remains a parameter in the Bayes factor and is formally separated from the prior probabilities of the individual tests. If these probabilities are also made equal to $p_{\Theta;i}$ then the EBF will lead to the same posterior probability as the two-groups model.

## Case studies

### Normal mean

The test of a normal mean against a point null is a benchmark in discussions of hypothesis testing. Recall the EBF for the two-sided test of $H_0:\mu = 0$ versus $H_1:\mu \neq 0$ (Eq 8)

$$EBF_{01} = \sqrt{2}\exp\left(-\frac{1}{2}(z^2 - 1)\right)$$

Here $z = x\sqrt{n}/\sigma$, where $x$ is the sample mean, $n$ the sample size and $\sigma$ the known standard deviation.

There is a correspondence between this EBF and the $P$-value, since $p = 2\Phi(-|z|)$. However, the EBF favours $H_0$ when $z^2 < 1 + \log 2$; thus in contrast to $P$-values and minimum Bayes factors, it can give evidence for either hypothesis. Indeed $p = 0.193$ when $z^2 = 1 + \log 2$, so any larger $P$-value corresponds to an EBF in favour of $H_0$ and smaller in favour of $H_1$. Fig 1 shows an almost linear relationship in log scale between EBF and $P$-value. This suggests a degree of resolution to the "irreconcilability of $P$-values and evidence" [42], for the EBF gives a Bayes factor interpretation to $Z$ and hence $P$. However, a given $P$-value corresponds to the same EBF irrespective of sample size, in contrast to ordinary Bayes factors [31, 48, 49]. The notion that a fixed $P$-value has less evidential value in larger samples is tied to the use of an informative prior for $H_1$. With prior ignorance about $H_1$, the posterior distribution is related to the $P$-value in such a way that the EBF has no further dependence on sample size.

Fig 1 also shows the BRC [23], in the present notation $\exp\left(-\frac{1}{2}(z^2 + 1)\right)$. This is also a posterior predictive approach, a key difference from EBF being that it takes replicate data under

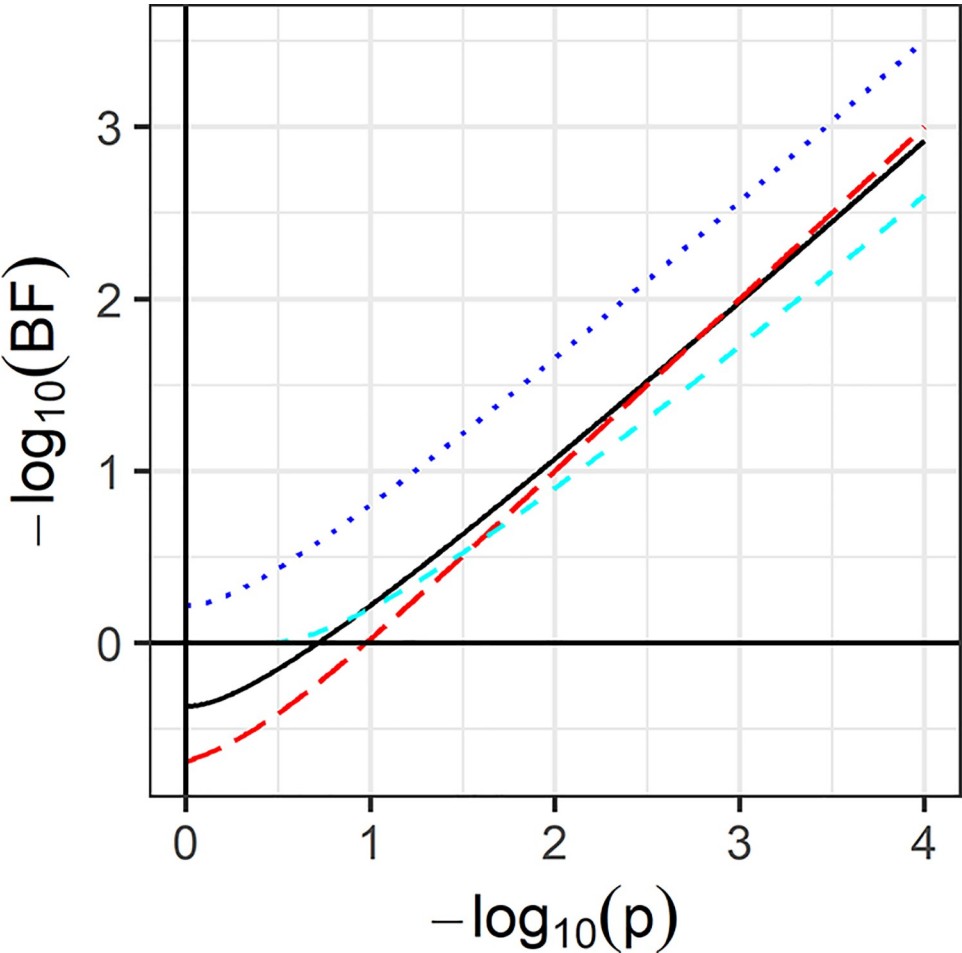

**Fig 1. Calibration of *P*-values for two-sided test of normal mean.** Solid black line, EBF for normal likelihood. Long dashed red line, *P*-value EBF with Beta likelihood. Dashed cyan line, Sellke et al. bound on the Bayes factor. Dotted blue line, Bayesian reference criterion. All quantities shown as negative logarithms with base 10.

the posterior distribution rather than the prior. Consequently the criterion always favours $H_1$, and does so more strongly than the minimum Bayes factor over all priors $\exp\left(-\frac{1}{2}z^2\right)$ [50, 51]. It is motivated by estimating Kullback-Leibler divergence rather than a Bayes factor, and is therefore less directly related to Bayesian hypothesis testing than the EBF. The figure also shows the *P*-value EBF which is in the same order of magnitude as the normal EBF. The approximation of $10p$ is clearly seen.

The evidence for $H_0$ is limited, even asymptotically, as $EBF_{01} = \sqrt{2e}$ when $z^2 = 0$. If $H_0$ does hold exactly then $Z^2$ follows a $\chi_1^2$ distribution regardless of sample size, with probability 0.193 of exceeding $1+\log 2$ and incorrectly providing evidence for $H_1$.

This lack of asymptotic consistency may be considered a shortcoming. This has been discussed for related quantities such as AIC, with common responses being that $H_0$ is never exactly true or that consistency is less critical for prediction models [52, 53]. Here I note that although a method may be asymptotically consistent, it may not be accurate at the current sample size. For if, in finite data, the Bayes factor favours a correct $H_0$ with high probability, then for small $\mu \neq 0$ it will incorrectly favour $H_0$ with almost the same probability. This is shown in Fig 2 for a fixed sample size of 1000. The EBF is compared to the unit information

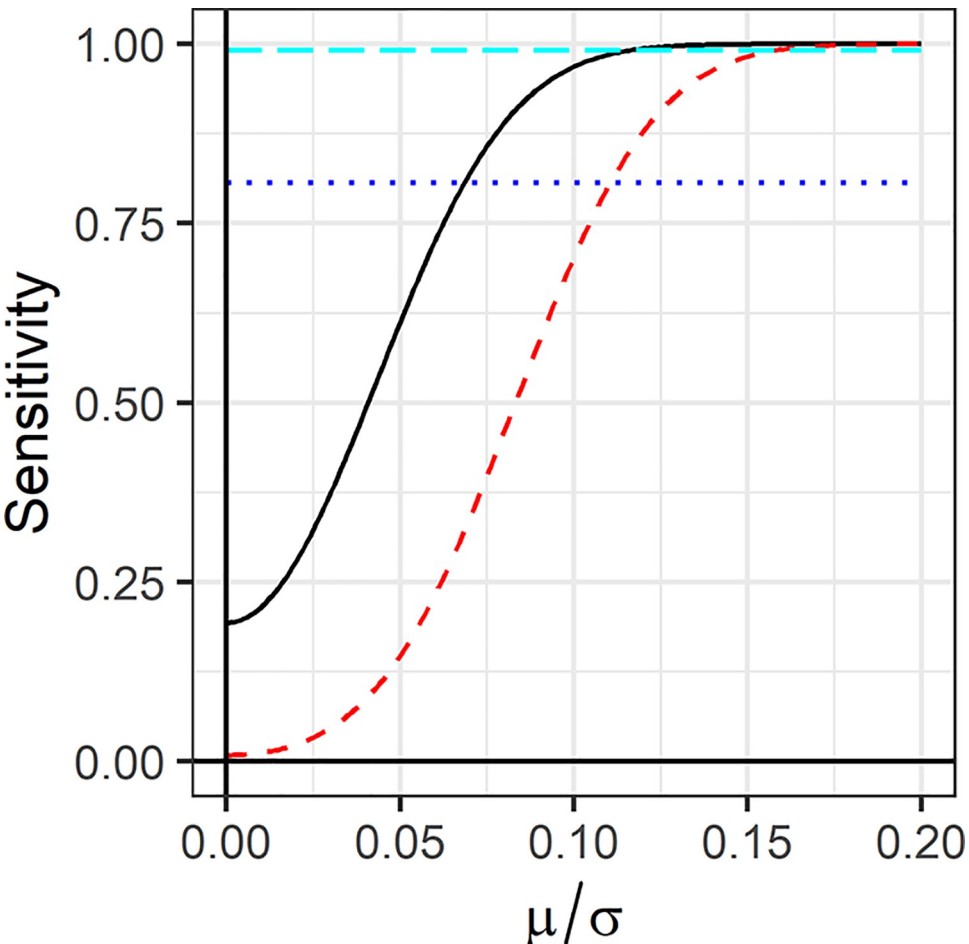

**Fig 2. Sensitivity of Bayes factors for two-sided test of normal mean.** Sensitivity is the probability that the Bayes factor favours the correct model, and is calculated from the $\chi^2$ distribution with non-centrality parameter $n\mu^2/\sigma^2$. Sample size is fixed to 1000. Solid black line, EBF as a function of $\mu/\sigma$. Dashed red line, unit information Bayes factor as a function of $\mu/\sigma$. Dotted blue line, EBF when $\mu = 0$. Long dashed cyan line, unit information Bayes factor when $\mu = 0$.

Bayes factor $\sqrt{n+1}\exp\left(-\frac{1}{2}z^2 n/(n+1)\right)$, which has the prior $N(0, \sigma^2)$ for $H_1$. This is regarded as minimally informative [12] and is approximately equivalent to the difference in Bayesian information (Schwarz) criterion for large $n$. For $\mu/\sigma \geq 0.15$ the unit information Bayes factor is nearly consistent at this sample size, but for small $\mu/\sigma$ it favours $H_1$ with low probability. As $\mu$ is unknown, asymptotic consistency may not be relevant to the data at hand.

The EBF has greater sensitivity against $H_1$ than the unit information Bayes factor, at the cost of fixed sensitivity against $H_0$. This contrast underlies the so-called Jeffreys-Lindley paradox [54, 55]. The dilemma for the practitioner is between consistency and sensitivity, with good arguments for both frequentist and Bayesian readings [56–58]. Here the behaviour of the EBF is more similar to frequentist testing, but this emerges from its objective construction rather than a philosophical position.

It is in fact remarkable that, with finite data, we can infer any evidence for the point $H_0$, compared to an infinitesimal departure from it, from a prior position of complete ignorance. The EBF has probability 0.807 of favouring $H_0$ when it is true, and $\log(EBF_{01})$ has expectation $\frac{1}{2}\log 2$, also favouring $H_0$. So in certain ways it is expected to favour a correct $H_0$, although never overwhelmingly. On the other hand, if data were to accrue without limit, one may as

well infer an informative prior from some finite subset and achieve consistent inference in the remainder. An EBF is only needed when data are finite and prior information is absent.

The same discussion applies to the one-sided test of $H_0:\mu = 0$ versus $H_1:\mu>0$ (Eqs 9, 10) and the $\chi^2$ test (Eq 12). However, for $H_0:\theta<0$ versus $H_1:\theta>0$ (Eq 11) the EBF depends only on the signed value of $z$ and will then be consistent for either hypothesis.

The two-sided EBF is approximately $2.33 \exp\left(-\frac{1}{2}z^2\right)$ which is a constant multiple of the minimum Bayes factor over all priors $\exp\left(-\frac{1}{2}z^2\right)$ [50, 51]. It is also similar to the minimum Bayes factor over all symmetric priors about 0, which is approximately $2 \exp\left(-\frac{1}{2}z^2\right)$ for $z>1.64$ [41]. These lower bounds, however, cannot favour $H_0$ and only give a limit on the evidence in favour of $H_1$; therefore they only quantify an absence of evidence and cannot support a conclusion favouring either hypothesis. The EBF by contrast is exact and can provide positive evidence for either hypothesis, albeit limited for $H_0$. The numerical similarity to the lower bounds is reassuring for the accuracy of those bounds. The generalisation of the EBF to multivariate and interval hypotheses, and to non-normal distributions, also goes beyond these previous results.

## Limited multiplicity

In the previous section an EBF was proposed for multiple testing, in which each test contributes a posterior distribution to the EBF for every other test. The contribution from a test to its own EBF is adjusted by the single-test correction (Eq 24). Clearly this reduces to the single-test EBF in the case of one test. Here I examine this heuristic EBF for a moderate number of tests.

Test of a normal mean were simulated with the number of tests $m$ varied from 1 to 10. In each case, three scenarios were simulated for the true means. Firstly, all means were set to zero. Secondly, the means were sampled from $N(0,1)$ independently in each simulated dataset. Thirdly, the means were fixed to a uniform grid on [−5, 5]. In each scenario, 10,000 sample means were simulated for each test assuming a population variance of 1 and a sample size of 100. With each sample mean, an independent replicate value was also simulated, to estimate the bias in each scenario.

Table 4 shows the bias in the log posterior marginal likelihood for $H_1:\mu\neq0$, with and without the proposed adjustment. As expected, the bias is $\frac{1}{2}$ for a single test and decreases towards zero as the number of tests increases. The rate of decrease depends on the true values of the means. The proposed adjustment does a good job of reducing the bias. Qualitatively similar results are observed for different sample sizes and models (not shown).

Table 5 compares the mean square error for the single- and multiple-test EBFs, where the error is the difference in log scale between the posterior marginal likelihood and the marginal likelihood with prior supplied from replicate data. The mean square error is seen to decrease in the multiple-test EBF as the number of tests increases. The decrease is greatest in scenario 1,

**Table 4. Bias in log posterior marginal likelihood for $m$ tests of a normal mean with $H_1:\mu\neq0$.** First column shows scenario numbers as described in the main text. Unadj, no adjustment for over-fitting. Adj, adjustment as proposed in Eq 24.

| | | $m = 1$ | 2 | 3 | 4 | 5 | 6 | 7 | 8 | 9 | 10 |
|---|---|---|---|---|---|---|---|---|---|---|---|
| 1 | Unadj | .493 | .250 | .166 | .122 | .099 | .083 | .071 | .063 | .054 | .050 |
| | Adj | -.007 | -.024 | -.021 | -.021 | -.016 | -.014 | -.012 | -.009 | -.010 | -.008 |
| 2 | Unadj | .504 | .500 | .490 | .445 | .466 | .297 | .230 | .332 | .283 | .241 |
| | Adj | .004 | -.000 | -.009 | -.025 | -.018 | -.038 | -.039 | -.044 | -.046 | -.038 |
| 3 | Unadj | .499 | .497 | .495 | .497 | .500 | .499 | .502 | .501 | .499 | .507 |
| | Adj | -.001 | -.003 | -.005 | -.003 | .000 | -.001 | .002 | .001 | -.001 | .007 |

**Table 5. Mean square error in log posterior marginal likelihood for *m* tests of a normal mean with $H_1:\mu\neq0$.** First column shows scenario numbers as described in the main text. Single, single-test EBF. Multiple, multiple-test EBF. Error is the difference between the log posterior marginal likelihood and the log prior marginal likelihood with prior from independent replicate data.

| | | $m=1$ | 2 | 3 | 4 | 5 | 6 | 7 | 8 | 9 | 10 |
|---|---|---|---|---|---|---|---|---|---|---|---|
| 1 | Single | .467 | .499 | .510 | .495 | .504 | .493 | .503 | .501 | .510 | .500 |
| | Multiple | .467 | .208 | .125 | .088 | .072 | .058 | .048 | .043 | .039 | .033 |
| 2 | Single | .508 | .499 | .490 | .484 | .507 | .495 | .498 | .513 | .510 | .489 |
| | Multiple | .508 | .499 | .470 | .401 | .421 | .267 | .200 | .306 | .247 | .223 |
| 3 | Single | .503 | .494 | .488 | .496 | .502 | .497 | .492 | .511 | .503 | .509 |
| | Multiple | .503 | .494 | .488 | .496 | .502 | .497 | .492 | .511 | .503 | .509 |

where all means are equal, whereas there is negligible decrease in scenario 3 where the means are well separated. Similar to the ODP, the EBF performs an adaptive shrinkage towards common effects where present. These results together suggest that the proposed EBF is useful for multiple testing situations.

## Large-scale multiplicity

In Fig 3, 1000 tests of a normal mean were simulated in which $\mu = 0$ for 900 tests and $\mu\sim N(0,1)$ for 100. The sampling variance is 1. The multiple test EBF shrinks the single test EBFs, reflecting reduced bias for each observed datum.

The lower line of circles shows the EBFs calculated with $p_{\Theta;i} = 1$ for all $i$, that is assuming $\mu\neq0$ for all tests. The results are virtually identical using the true value of $p_{\Theta;i} = 0.1$ (not shown). The upper line of crosses shows the EBFs still similar with $p_{\Theta;i} = 0.01$. In this case a default of $p_{\Theta;i} = 1$ would not be misleading when the truth is unknown.

Note that more extreme EBFs can occur as the number of tests increases, and so the usual issues of multiple testing remain. Although the multiple-test EBFs are reduced compared to the single-test version, there is little improvement in the ranking of the tests: the non-zero means have a mean rank of 411.8 for the single-test EBF and 411.28 for the multiple-test, where lower ranks indicate greater evidence against $H_0$. However, the multiple-test EBF selects a higher proportion of true positives over a range of thresholds (Fig 4), in line with the ODP theory.

## Qualitative interpretation of Bayes factors

Often the order of magnitude of the Bayes factor is of greater interest than its precise value. Several authors have proposed interpretations on the log scale with descriptions such as "moderate" and "strong" evidence. Jeffreys [9] used base 10, with Lee and Wagenmakers [30] and Held and Ott [31] suggesting minor adjustments to his scale. Kass and Raftery [1] used natural logs, while Royall [32] used base 2 for simple likelihood ratios. However, all authors used subjective experience to identify significant points on their scales, and it is unclear which scale is most appropriate for general use.

Here I propose measuring Bayes factors in logarithmic units with base 3.73. Briefly, the argument is the following, with further details in the supplementary material. Define weaker belief to be that which is more easily modified by evidence, and stronger belief to be that which is less easily modified. From Bayes' theorem, the effect of evidence on belief is the first derivative of the logistic function. The sharpest distinction between weaker and stronger belief occurs where the effect of evidence is changing most rapidly. This is where the third derivative is zero, which is $\pm\log\left(\frac{\sqrt{3}+1}{\sqrt{3}-1}\right)$. With that boundary, a Bayes factor of $\frac{\sqrt{3}+1}{\sqrt{3}-1} = 3.73$ updates any weaker belief to a stronger belief and is said to comprise one unit of evidence.

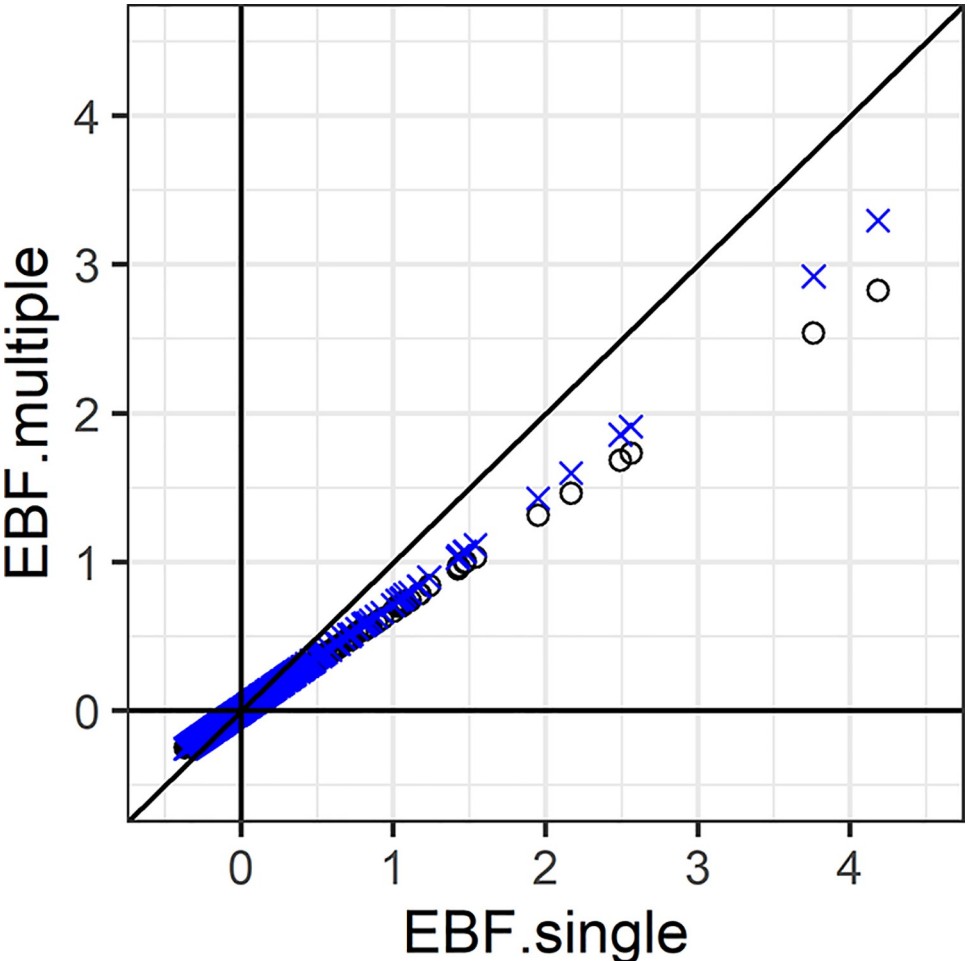

**Fig 3. Single- and multiple-test EBFs.** EBFs are in favour of $H_1$:$\mu \neq 0$, for 900 simulated tests with $\mu = 0$ and 100 with $\mu \sim N(0,1)$. Black circles, multiple-test EBF with $p_{\Theta;i} = 1$. Blue crosses, multiple-test EBF with $p_{\Theta;i} = 0.01$. The true value is $p_{\Theta;i} = 0.1$. EBFs shown in $\log_{10}$ scale.

For the normal theory and $P$-value EBFs, the $P$-values corresponding to the first few units of evidence are shown in Table 6. These are remarkably similar to traditional significance levels. The EBF and the proposed units together give an evidential account of classical testing, whereby $P = 0.05$ is roughly enough evidence to decrease the qualitative belief in $H_0$, $P = 0.01$ to decrease it further and $P = 0.005$ further again. Recall also that against even prior odds, the $P$-value EBF for $P = 0.005$ yields posterior probability of 4.8% for $H_0$ (Eq 22), supporting the proposal for this level as a general threshold for hypothesis testing [43].

The EBF and the proposed units give some rationale to traditions that have had only intuitive justification. Although this does little to discourage continued use of $P$-values, I prefer the EBF as the exact calibration is different for each test, and it has a direct evidential interpretation as in the following examples.

## Examples

Given the calibrations of EBFs to $P$-values, the main operational difference is in interpretation. Here are two examples in which the interpretation of $P$-values has fuelled discussions of

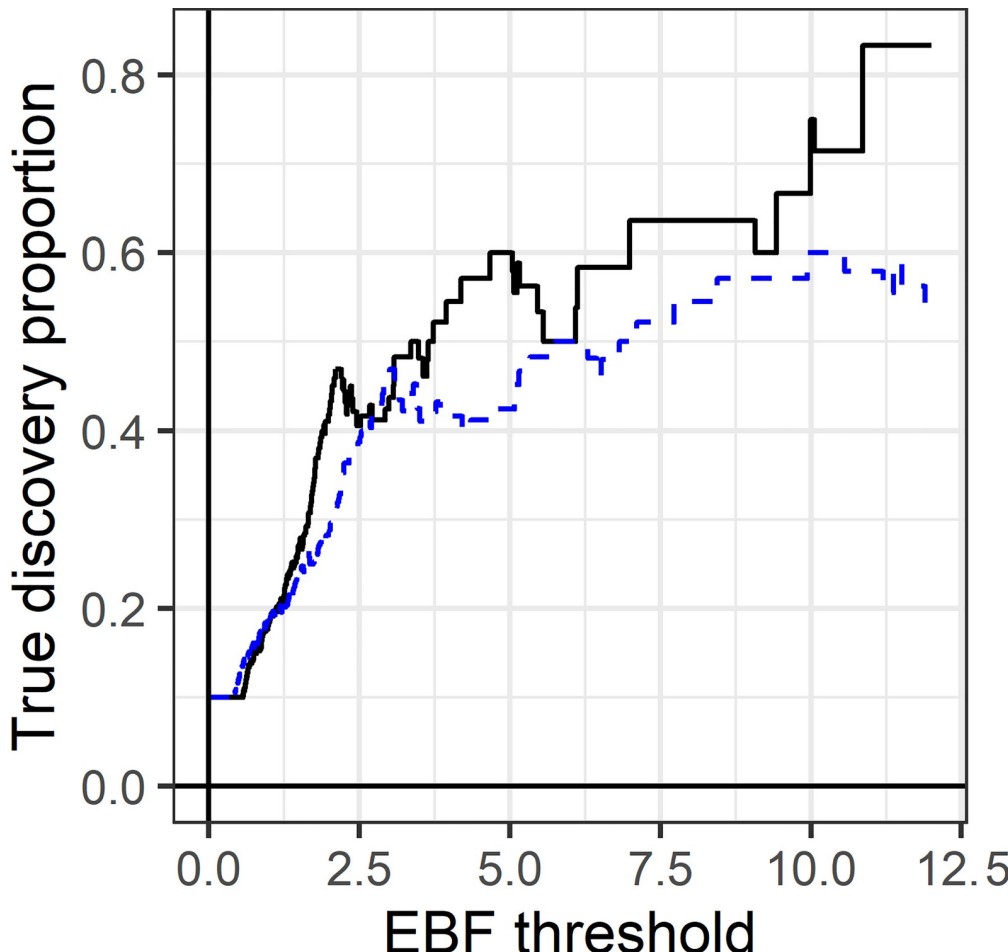

**Fig 4. Proportion of true positives in Fig 3 among all tests with EBF above a threshold.** Solid black line, multiple-test EBF. Dashed blue line, single-test EBF.

inference. In both cases there is little relevant prior information for an ordinary Bayes factor. The EBF seems to provide a direct and transparent interpretation.

The first concerns the use of stents to treat patients with angina [59]. In the first blinded randomized controlled trial of their effectiveness, patients receiving a stent increased their exercise time by 16.6s more than the controls [60]. With standard error 12.96s, $z = 1.28$ and two-sided $P = 0.2$, the authors concluded that stents provided no clinical benefit, and discussion ensued on whether clinical guidelines should be revised [61]. As a non-significant $P$-value is not evidence for $H_0$, such discussion appeared premature [59].

**Table 6. $P$-value calibrations for EBFs corresponding to the first four units of evidence.** Normal mean and $\chi^2$ EBFs from Eqs (8) and (12). $P$-value EBF from the $10p$ calibration (Eq 22). EBFs shown in favour of $H_1$. The table shows, for example, that for the test of a normal mean a $P$-value of 0.038 corresponds to an EBF of one unit in favour of the posterior distribution.

| Units | EBF | Normal mean (2-sided) | $\chi^2$ (2df) | $\chi^2$ (3df) | $P$−value |
|---|---|---|---|---|---|
| 1.0 | 3.73 | 0.038 | 0.049 | 0.052 | 0.027 |
| 2.0 | 13.9 | 0.008 | 0.013 | 0.016 | 0.007 |
| 3.0 | 52.0 | 0.002 | 0.004 | 0.005 | 0.002 |
| 4.0 | 194 | 0.0005 | 0.001 | 0.001 | 0.0005 |

In this case the two-sided EBF (Eq 8) is 1.03, favouring no effect but minimally. A simple interpretation is that the study gives no reason to change existing views on whether stents give any benefit. For a more nuanced interpretation, the authors considered an increase of at least 30s to be a clinically meaningful effect, but the study was only powered to detect an effect of at least 34s. The EBF comparing $H_0$:$[-\infty, 30)$ to $H_1$:$[30, \infty]$ is 2.29 (Eqs 1 and 5), supporting the hypothesis of no clinically meaningful benefit, though at less than one unit, not to a degree that demands revision of clinical guidelines.

The second example is the identification of the Higgs boson from particle collisions [62, 63]. In simple terms, the production of a new particle was inferred from a count of events $5\sigma$ above background expectation (one-sided $P = 3\times10^{-7}$). The Standard Model predicts an excess of $5.8\sigma$ for the Higgs boson and the observations were thus deemed compatible with its presence and incompatible with its absence.

The substantial media interest brought the usual clumsiness in communicating $P$-values, such as "a 0.00003% probability that the result is due to chance" [64]. For a normal likelihood, the one-sided EBF (Eq 9) for $5\sigma$ is $1.48\times10^5$ in favour of a new particle; against even prior odds this gives 0.0007% posterior probability of a chance result. We may also calculate the EBF for the Standard Model expectation of $5.8\sigma$ versus a two-sided alternative, giving 1.69 in favour of the Higgs boson. Together these EBFs give positive evidence that a new particle was produced and that it is the Higgs boson as opposed to some other phenomenon.

## Discussion

The present aim is to construct objective Bayes factors with subjectively reasonable priors. Although the approach is motivated by Bayesian learning, the result is in some ways closer to frequentist testing. For tests of a point null there is a fixed asymptotic distribution under $H_0$, allowing control of type-1 error. Like the $P$-value, the EBF appeals to hypothetical replicate data, and thus violates the likelihood principle, although uncertainty about the sampling model may be mitigated by Bayesian model averaging. EBFs can be calibrated to $P$-values; while the exact calibration is different for each test, a general calibration of $10p$ has been proposed (Eq 22).

Several other calibrations of $P$-values to Bayes factors have been proposed previously, the most well-known being $-e\,p\,log(p)$ [40, 41, 65, 66]. These are all lower bounds on the Bayes factor in favour of $H_0$, but are constrained to be at most 1. Consequently, these calibrations cannot produce positive evidence in favour of either hypothesis; at best they say that there is no more than so much evidence against $H_0$. As such, they only quantify an absence of evidence and are not very useful for decision making. In contrast the EBF calibrations are exact, and can favour either hypothesis. They can be viewed as giving a Bayes factor interpretation to the $P$-value, alleviating some concerns about frequentist testing [5]. Informally, the EBF suggests that for $P$-values near 1 one may increase belief in $H_0$, but only to a limited extent since there are also realisations of $H_1$ under which such $P$-values are possible.

Such an interpretation may seem surprising, since the $P$-value quantifies evidence against but not for $H_0$. However, the $P$-value is just a function of the data, and can be used to obtain a Bayes factor if $H_1$ is appropriately specified, here as the posterior distribution. Thus the EBF presents a Bayesian/frequentist compromise in that it may be used as a frequentist statistic, having a fixed distribution under a point $H_0$, while a $P$-value may be calibrated to a Bayes factor in which $H_1$ is the posterior distribution.

This seems reasonable in parametric tests, in which a $P$-value is often supplemented with a point estimate or confidence interval. Then the EBF similarly reports both the posterior distribution and the evidence supporting it. But in non-parametric tests the correspondence

between $P$ and EBF is more tenuous. A calibration of $10p$ is obtained by assuming a parametric model for $P$, comparing the uniform distribution to $Beta(1, \beta)$. Similarly for goodness-of-fit tests, Eq (12) might be used to obtain an EBF comparing the central $\chi^2$ distribution to a non-central distribution. In these cases the EBF requires a model for the alternative that is absent in the classical test, and the Bayesian interpretation of $P$ should be treated with caution.

From the Bayesian perspective there are two main issues. The first is the lack of asymptotic consistency when testing a point null embedded within a composite alternative. I have argued that this is less important in the finite data scenarios for which EBFs are intended. Indeed it is remarkable that, in finite data, the EBF is expected to favour a true point null over an infinitesimal departure. The inconsistency of the EBF is not a defining characteristic, but a property that emerges from its objective construction.

The second issue concerns coherence. While the EBF is on the same scale as a proper Bayes factor, its prior does not precede the data and there is no formal justification for updating prior odds by Bayes' theorem [67]. Of course this may be said of any empirical Bayes method, and is only brought into relief by the single-test EBF. If empirical Bayes is acceptable for multiple testing, but not for a single test, one must ask where the difference lies since a bias is present in all cases. A pragmatic line is the following: if the data at hand would yield this Bayes factor given replicate data, then one may as well update the prior odds now. Such an argument is not strictly Bayesian, but is a compromise combining the principle of Bayesian updating with the frequentist idea of a replicate experiment.

In reality, literal applications of Bayesian updating or frequentist sampling are uncommon. Especially in research contexts, the Bayesian and frequentist constructions merely serve as rhetorical devices to enable inference from the single case. My position is closer to the evidential view, according to which the primary aim is to quantify the evidence in data [32, 34, 68]. That evidence can then be interpreted within one paradigm or other according to the agent's subjective will [69], but the evidence stands itself as an objective property of the data. At the community level, inference proceeds by consensus based on multiple lines of evidence, and formal testing is less important than the evidence base [70]. From this perspective the EBF is attractive as it admits Bayesian, frequentist and (via asymptotic equivalence to WAIC, Eq (7)) information interpretations.

Taken purely as a measure of evidence, the EBF is one of several possible approaches including information criteria [34], maximised likelihood ratios [71, 72] and frequentist error rates [33]. Indeed the posterior Bayes factor could simply be taken at face value (rejoinder of Aitkin [21]). These quantities are each on an absolute scale, and a key question is how to interpret a given value. I have given arguments for interpreting Bayes factors with logarithmic base 3.73, an approach that is lacking from other accounts of evidence. The availability of such a scale is an advantage of the Bayes factor, for it allows a more objective interpretation of the strength of evidence. Without this scale, there is little reason to adjust the posterior Bayes factor to the EBF. The $P$-value calibrations for EBFs on this scale are similar to traditional significance levels (Table 6), so that this framework is consistent with established practice while providing an arguably more transparent interpretation.

The use of test-based Bayes factors avoids some of the problems associated with nuisance parameters and likelihood-free approaches. Where, for example, an estimating equation is used whose solution is asymptotically normal, one may simply take the estimator to the normal EBF. However a potential limitation is the need for a prior predictive distribution for replicate data. In the tests considered in this paper, the expected bias either does not depend on the prior predictive, or there is a natural choice as in the binomial. But in other cases there may not be a clear model for replicate data and an EBF may be hard to define.

I have focused on some hypothesis tests that are commonly applied in routine fashion. These are the focus of much of the debate around testing, but the EBF is more generally applicable, for example to multi-dimensional parameters, non-nested and interval hypotheses. Similar approaches to those described in this paper can be used to develop EBFs in a wider range of settings.

## Supporting information

**S1 Text.**
(DOCX)

## Author Contributions

**Conceptualization:** Frank Dudbridge.

**Formal analysis:** Frank Dudbridge.

**Funding acquisition:** Frank Dudbridge.

**Investigation:** Frank Dudbridge.

**Methodology:** Frank Dudbridge.

**Software:** Frank Dudbridge.

**Writing – original draft:** Frank Dudbridge.

**Writing – review & editing:** Frank Dudbridge.

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
