## [Decision Letter · Decision Letter 0]

21 Jul 2023

PONE-D-23-08009Empirical Bayes factors for common hypothesis tests

PLOS ONE

Dear Dr. Dudbridge,

Thank you for submitting your manuscript to PLOS ONE. After careful consideration, we feel that it has merit but does not fully meet PLOS ONE’s publication criteria as it currently stands. Therefore, we invite you to submit a revised version of the manuscript that addresses the points raised during the review process.

One of the reviewers is pointing out concerns regarding the manuscript, mainly regarding the justification of some steps in the statistical analysis but also in the conclusions. I suggest you perform a major revision to better highlight the contribution of your work and the rigurosity of the developed methodology. If you feel that you are able to comprehensively address the reviewers’ concerns, please provide a point-by-point response to these comments along with your revision. Please show all changes in the manuscript text file with track changes or colour highlighting. If you are unable to address specific reviewer requests or find any points invalid, please explain why in the point-by-point response. Please submit your revised manuscript by Sep 04 2023 11:59PM. If you will need more time than this to complete your revisions, please reply to this message or contact the journal office at plosone@plos.org. Please include the following items when submitting your revised manuscript:A rebuttal letter that responds to each point raised by the academic editor and reviewer(s). You should upload this letter as a separate file labeled 'Response to Reviewers'.A marked-up copy of your manuscript that highlights changes made to the original version. You should upload this as a separate file labeled 'Revised Manuscript with Track Changes'.An unmarked version of your revised paper without tracked changes. You should upload this as a separate file labeled 'Manuscript'.

We look forward to receiving your revised manuscript.

Kind regards,

Pablo Martin Rodriguez

Academic Editor

PLOS ONE

Journal Requirements:

Reviewers' comments:

Reviewer's Responses to Questions

**Comments to the Author**

1. Is the manuscript technically sound, and do the data support the conclusions?

Reviewer #1: Partly

Reviewer #2: Yes

2. Has the statistical analysis been performed appropriately and rigorously? 

Reviewer #1: N/A

Reviewer #2: Yes

3. Have the authors made all data underlying the findings in their manuscript fully available?

Reviewer #1: Yes

Reviewer #2: Yes

4. Is the manuscript presented in an intelligible fashion and written in standard English?

Reviewer #1: Yes

Reviewer #2: Yes

5. Review Comments to the Author

Reviewer #1: Under review is the proposal for an "empirical Bayes factor". The derivation begins by considering the "bias" in the posterior marginal likelihood, which the empirical Bayes factor involves correcting for. It was unclear to me why the formula for b_H(x) was designated as "bias" that should be corrected. It was not clear to me why the averaged behavior across draws of Y should be of first-order importance in a Bayesian analysis. It seems quite standard in Bayesian analysis for quantities to be biased (as generally Bayesian estimates are biased) and it would be unclear why in general such bias would need to be corrected for. It would seem necessary to set up a formal justification for this, and other steps in the statistical analysis. Also, when it comes to the concrete application to the normal mean in section 4.1, it seems the answer is approximately 2.33 exp(-z^2/2), which essentially is a constant multiple of the standard minimum Bayes factor which is exp(-z^2/2) and very close to the minimum Bayes factor in equation 10 of Held and Ott which is approximately 2 exp(-z^2/2). Despite the numerical similarity, the proposal under review seems to suggest a dramatic difference between the calculations here and the standard results, which I do not really see. Overall, at least to me, the proposal does not make a sufficient case for the statistical approach taken here.

Reviewer #2: The article proposes an empirical Bayes factor in order to obtain a measure of evidence in favor of a hypothesis to be tested. In his approach, the data are used in the construction of the priori with the justification that, although the data are used twice, the data used in the posteriori would be obtained from a replication of the experiment.

The author defends the use of his proposal, highlighting the interpretability of the proposed empirical Bayes factor.

The text presents the formulation of this proposed measure of evidence and how it is expressed in tests of some distributions, such as normal, t, binomial, F and non-parametric tests, and also in multiple tests. Furthermore, it compares for some situations the relationship between the P-value and the empirical Bayes factor. Additionally, practical applications are presented, exemplifying that the empirical Bayes factor presents adequate conclusions to the tested hypotheses. The author ends the article assuming that the proposal is not purely Bayesian, but that the objective was just to obtain a good measure of evidence in favor of a hypothesis.

In my opinion, the proposed measure is a creative alternative for calculating the evidence in favor of a hypothesis, in addition to correcting inconsistencies that can be found when using the traditional Bayes factor, without correcting the bias. In the supplementary material, the author explains how to interpret and use this measure (even without the supplementary material it is more difficult to understand how to consider and interpret the result obtained in the calculation of the proposed measure) and shows that, although there is no improvement to replace a value- P, the results can be considered a little more interpretable. The work even includes a package in R that was developed to calculate the proposed measure. My recommendation is to accept the submission.

6. PLOS authors have the option to publish the peer review history of their article (what does this mean?). If published, this will include your full peer review and any attached files.

Reviewer #1: No

Reviewer #2: No

---

## [Author Response · Author response to Decision Letter 0]

26 Jul 2023

A detailed response is provided in the attached document.

---

## [Decision Letter · Decision Letter 1]

23 Oct 2023

PONE-D-23-08009R1Empirical Bayes factors for common hypothesis testsPLOS ONE

Dear Dr. Dudbridge,

Thank you for submitting your manuscript to PLOS ONE. After careful consideration, we feel that it has merit but still does not fully meet PLOS ONE’s publication criteria as it currently stands. Therefore, we invite you to submit a revised version of the manuscript that addresses the points raised during the review process. You will see that the reviewer has some concerns regarding your method. I believe that your contribution will be better presented after answering these new comments.

We look forward to receiving your revised manuscript.

Kind regards,

Pablo Martin Rodriguez

Academic Editor

PLOS ONE

Reviewers' comments:

Reviewer's Responses to Questions

**Comments to the Author**

1. If the authors have adequately addressed your comments raised in a previous round of review and you feel that this manuscript is now acceptable for publication, you may indicate that here to bypass the “Comments to the Author” section, enter your conflict of interest statement in the “Confidential to Editor” section, and submit your "Accept" recommendation.

Reviewer #3: (No Response)

2. Is the manuscript technically sound, and do the data support the conclusions?

Reviewer #3: No

3. Has the statistical analysis been performed appropriately and rigorously? 

Reviewer #3: I Don't Know

4. Have the authors made all data underlying the findings in their manuscript fully available?

Reviewer #3: Yes

5. Is the manuscript presented in an intelligible fashion and written in standard English?

Reviewer #3: Yes

6. Review Comments to the Author

Reviewer #3: - Overall: The paper was well referenced, and your main motivation and derivation for the empirical Bayes factor appears sound, and I could not see any problems with the mathematical details. My main concern is that you have oversold your method. In many places I think your claims for the empirical Bayes factors are too broad. In other words, the EBF cannot do as much as you claim it can. See my detailed comments, especially with respect to lines 60-61 and 81-83 and 87-89 and Section 2.5.

- lines 60-61: you state "This allow[s] standard routine analyses to proceed with minimal change, essentially replacing the final calculation of a P-value with an EBF". This is confusing to me because the EBF is acting like a Bayes factor which is measuring something different from a P-value. P-values relate to rejecting the null hypothesis (and frequentists treat the null and alternative hypotheses in fundamentally a different manner...you cannot reject the alternative hypothesis with large p-values), Bayes factors are interpreted as in line 97 (and Bayes factors treat the hypotheses similarly, so that BF01=1/BF10). So I would delete this sentence.

- Line 64: you state: "the EBF is equivalent to the difference in WAIC under a vague prior". It is the difference between WAIC and what?

- lines 81-83: you talk about strength of evidence for qualitatively describing Bayes factors (like Table 1 of the supplement), and then you talk about "Interpretations have been proposed with base 10...". Finally, stating "But as these scale are subjective, they do not of themselves warrant the Bayes factor interpretation." I do not understand the last sentence. All of the qualitative interpretations of Bayes factors (as in Table 1, Supplement) are subjective, but they are still a "Bayes factor interpretation", albeit an interpretation based on somewhat arbitrary descriptions. I do not understand why your interpretation of Bayes factors with base 3.73 is much different than the interpretations of log10, natural log, or log2. It is just an argument for using base 3.73 instead of 10,e, or 2. Unless I am misunderstanding, then the real impact is another column with new rows to be added to Supplementary Table 1 with your terminology (see Suppl. Table 3) at the appropriate likelihood ratio values (as in Suppl. Table 4), for example, LR=3.73 gives 'weaker disbelief', LR=13.9 gives 'stronger disbelief', etc. Is that right? Your qualitative interpretations seem reasonable, and your derivation of the 3.73 scale is interesting, but I just do not see yours as "objective" and the previously proposed interpretations as "subjective". In my mind, all are somewhat arbitrary. Similarly, rejecting at p<=0.05 is also somewhat arbitrary and is just a convention.

- lines 87-89 (see also lines 25-26): You state "We will see that this framework yields a surprising re-interpretation of classifcal significance testing, realizing a compromise between Bayesian and frequentist paradigms." I think this overstates what you are doing. You have a way to estimate Bayes factors with a more objective prior, but it is still a Bayes factor, which is measuring something different from a P-value. It is not doing classical significance testing, which only rejects the null hypothesis or fails to reject the null hypothesis (since large P-values are not necessarily evidence for the alternative, they could just be a lack of evidence against the null hypothesis). For a recent Bayesian interpretation of frequentist inferences, see e.g., Fay, Proschan, Brittain, and Tiwari (2022) Statistical Science 37(4): 455-472.

-line 105: can you state the theta belongs in the set Theta_H, i.e., use "..for theta in Theta_H in light of x." (using the appropriate math symbols for 'theta in Theta_H').

-lines 204,207,210: typo? theta should be mu? For example, "H1: theta > 0" should be "H1: mu >0" ?

- line 218-219: "Equation (12) can therefore be used for the usual tests of contingency tables and goodness of fit." I do not understand this sentence. Usualy a goodness of fit test is calculated as a "pure" significance test (see e.g., Cox and Hinkley, 1974, Theoretical Statistics, Chapman & Hall), where the alternative hypothesis is not explicitly defined. Please either give more details or delete that sentence.

- line 226-7: you state "This formulation accommodates the classical one- and two-sample test tests for the means,...." Are these one-sided tests or tests of the null that mu=0? Please explicitly state the two hypotheses. If it is H0: mu=0 vs H1: mu ne 0, then how do you integrate over Theta_H0?

-line 262: You give the expected bias in Table 2 when Theta_H=[0,1]. I am not quite sure how Table 2 applies to standard hypotheses such as H0: p <= 0.5 vs H1: p > 0.5. Please give an example like that so the reader has more of an idea how it applies in practice.

-lines 267-8: You state "This dependence on the sampling model is a departure from the likelihood principle,...". But even using Jeffreys prior with the usual Bayes factor on the binomial and negative binomial distributions is a departure from the likelihood principle, because it gives different priors for the two distributions (see e.g., Robert, 2007, The Bayesian Choice, Springer, p. 132). Please rewrite, otherwise it makes it seem like EBF is uniquely special in this sense.

- Section 2.5: One of my primary concerns with this paper is best explained with this section. Suppose you have a permutation test p-value. The p-value is calculated under exchangeability assumptions that hold under the null hypothesis but not under the alternative. For example, a Wilcoxon-Mann-Whitney test is like that. Then there is a fundamental difference between the null hypothesis (p-value is interpreted under the assumptions of the null hypothesis) and the alternative hypothesis (p-value is not interpreted under the assumptions of the alternative hypothesis). So the null and alternative hypotheses are treated differently. Small p-values reject the null but large p-values do not reject the alternative! But for Bayes factors the two hypotheses are treated similarly, since BF01 is equal to 1/BF10. So there is a fundamental difference between p-values and Bayes factors. You can talk about the relationship between the two (see e.g., Held and Ott, 2018, ref 41), but you will have to be much more careful and precise about the relationship than you have been in your paper. I am afraid readers will misunderstand the interpretation of EBF applied to nonparametric test. In fact, I cannot see how the EBF can be precisely and properly interpreted for the nonparametric P-value case. It would help me if you could go through the details of a simple example of a nonparametric p-value (e.g., Wilcoxon-Mann-Whitney test), write out the two hypotheses, and interpret the EBF (for both EBF_01 and EBF_10=1/EBF_01).

- lines 359-360: Please define H in "given that H is true for test i". Since theta_i is different for each i, I assume the H must be different for each i. If not, please explain. Please give more details.

- line 411: "The test of a normal mean against..." Please give the details of the notation. If z=x/sigma, what is x (the mean?), and what is sigma (the known standard deviation?), and how does the sample size affect EBF and the P-value? Please have a sample size of n>1, since it is important to see how the EBF and P-values change with sample size.

- line 414: you state: "There is clearly a correspondence between this EBF and the P-value." Please be kind to the reader and write out the P-value using the same notation as the EBF on line 413. How do they relate?

- line 415-17: you state: "..thus in contrast to P-values and minimum Bayes factors, [EBF] can give evidence for either hypothesis...". I do not understand how "Fig. 1 shows an almost linear relationship in log scale between EBF and P-value" yet somehow EBF can give evidence for either hypothesis but the P-value cannot. If you have that relationship, then you can just multiply the P-value by a constant to approximately get the EBF. Are you saying that we can extract a Bayesian interpretation from the P-value by multiplying it by a constant in this case?

- line 420-421: "..a given P-value corresponds to the same EBF irrespective of sample size,.." Perhaps if you include the sample size in this section it might help the reader understand why this is so.

-line 444-445: "...infinite sample sizes never occur..." That is not the point of consistency. The point of consistency is that as sample sizes get larger and larger, you get closer to the right answer. With inconsistent methods even as the sample size gets very large, it does not approach the right answer.

-lines 655: "..existence of such as scale is crucial...without it there is no way to objectively interpret the strength of evidence" I think this statement oversells the "objectivity" of the scale. I think it is just another scale (see my earlier comments about lines 81-83).

- Supplement line 42-43: you state "The following proposal relates the evidence to its effect on probability, independently of the prior odds." This seems to imply that the previous proposals (Table 1, Supplement) were not independent of the prior odds. But all of Suppl. Table 1 is for Bayes factors that are independent of the prior odds. Perhaps rewrite that sentence since your proposal is not that different from those of Suppl. Table 1.

7. PLOS authors have the option to publish the peer review history of their article (what does this mean?). If published, this will include your full peer review and any attached files.

Reviewer #3: No

---

## [Author Response · Author response to Decision Letter 1]

20 Nov 2023

Responses are in the attached document.

---

## [Decision Letter · Decision Letter 2]

15 Jan 2024

Empirical Bayes factors for common hypothesis tests

PONE-D-23-08009R2

Dear Dr. Dudbridge,

We’re pleased to inform you that your manuscript has been judged scientifically suitable for publication and will be formally accepted for publication once it meets all outstanding technical requirements.

Kind regards,

Pablo Martin Rodriguez

Academic Editor

PLOS ONE

Additional Editor Comments (optional):

Reviewers' comments:

Reviewer's Responses to Questions

**Comments to the Author**

1. If the authors have adequately addressed your comments raised in a previous round of review and you feel that this manuscript is now acceptable for publication, you may indicate that here to bypass the “Comments to the Author” section, enter your conflict of interest statement in the “Confidential to Editor” section, and submit your "Accept" recommendation.

Reviewer #3: All comments have been addressed

2. Is the manuscript technically sound, and do the data support the conclusions?

Reviewer #3: (No Response)

3. Has the statistical analysis been performed appropriately and rigorously? 

Reviewer #3: (No Response)

4. Have the authors made all data underlying the findings in their manuscript fully available?

Reviewer #3: (No Response)

5. Is the manuscript presented in an intelligible fashion and written in standard English?

Reviewer #3: (No Response)

6. Review Comments to the Author

Reviewer #3: (No Response)

7. PLOS authors have the option to publish the peer review history of their article (what does this mean?). If published, this will include your full peer review and any attached files.

Reviewer #3: No

---

## [Editor Report · Acceptance letter]

5 Feb 2024

PONE-D-23-08009R2 

PLOS ONE

Dear Dr. Dudbridge, 

I'm pleased to inform you that your manuscript has been deemed suitable for publication in PLOS ONE. Congratulations! Your manuscript is now being handed over to our production team.

Kind regards, 

on behalf of

Professor Pablo Martin Rodriguez 

Academic Editor

PLOS ONE